# Effectiveness of a Modified Administration Protocol for the Medical Treatment of Feline Pyometra

**DOI:** 10.3390/vetsci9100517

**Published:** 2022-09-22

**Authors:** Simona Attard, Roberta Bucci, Salvatore Parrillo, Maria Carmela Pisu

**Affiliations:** 1Palermovet—Centro Diagnostico Veterinario, 90125 Palermo, Italy; 2Animals Theriogenology Service, Veterinary Teaching Hospital, Faculty of Veterinary Medicine, University of Teramo, Piano d’Accio, 64100 Teramo, Italy; 3VRC—Centro di Referenza Veterinario, 10138 Torino, Italy

**Keywords:** cat, pyometra, modified treatment, aglepristone

## Abstract

**Simple Summary:**

Pyometra is a common reproductive disorder traditionally managed by ovariohysterectomy, although in selected cases (such as breeding subjects or patients with anesthesiologic risk) medical treatment is preferable. This paper aimed to describe the effectiveness of a modified aglepristone administration protocol. Five intact queens were referred for pyometra and were treated with 15 mg/kg of aglepristone on Day 0 (D0), D2, D5, and D8, along with antibiotic treatment (marbofloxacine, 3 mg/kg). Regular ultrasonographic exams confirmed the complete resolution of the infection. After treatment, 3 out of 4 mated cats had an uneventful pregnancy. The results obtained, although limited to a small group, are promising. Further studies are planned to verify its effectiveness in the long-term prevention of recurrence.

**Abstract:**

Pyometra is a common uterine disease of dogs and cats, typical of the luteal phase. Traditionally, ovariohysterectomy was considered the elective treatment for pyometra, but in some cases, such as breeding subjects or patients with a high anesthesiologic risk, medical treatment is preferred. Aglepristone is a progesterone receptor blocker and its use proved to be effective for the medical treatment of pyometra in bitches and queens. The aim of this work is to report the effectiveness, in the feline species, of a modified aglepristone administration protocol. Five intact queens were referred to veterinary care centers for pyometra. Aglepristone (15 mg/kg) was administered at D0, D2, D5, and D8, as described by Contri and collaborators for dogs. An antibiotic treatment (marbofloxacin, 3 mg/kg) was associated, and uterine conditions were checked with regular ultrasonographic exams. The uterus returned to its normal condition 10 days after starting the treatment and no adverse effects were reported. After treatment, three queens had an uneventful pregnancy. Even if the treated group was restricted and homogeneous, the proposed modified protocol proved to be useful and promising for the medical treatment of pyometra in cats; further studies are planned to verify its effectiveness in the long-term prevention of recurrence.

## 1. Introduction

Pyometra is a common illness that affects intact female dogs and cats [1] and rarely other small animals, such as hamsters [2] and rabbits [3]. The disease is characterized by a suppurative bacterial infection of the uterus, typically occurring during the luteal phase [1].

Affected patients are generally intact cats over 4–7 years old [4], with a range of 1–20 years [5,6,7,8,9].

Although its etiopathogenesis is not completely understood, it is believed that cystic endometrial hyperplasia (CEH) can predispose to pyometra [10]; nevertheless, the two disorders may develop independently [1,11]. When the lumen of the hyperplastic endometrium is filled with fluid, an opportunistic bacterial infection may develop, mainly caused by *E. coli* [12].

Clinical signs of pyometra in cats are generally absent or mild and mostly non-specific, with anorexia and lethargy being the most common [8,9,13]. Other signs have been reported, such as hyperthermia and vomiting [7], and, in severe cases, also sepsis [14], peritonitis and death [9]. Stanley et al. [15] even reported a uterine torsion following a pyometra. Unlike dogs, polyuria-polydipsia are generally absent in the feline species [8], while vaginal discharge is usually present, due to poor cervical sealing, although often it may not be evident due to cat cleaning habits [1].

Changes in hematological and biochemical profiles are also reported in the literature, such as leukocytosis with left shift [10], and Vilhena et al. [16] also detected Acute Phase Proteins and biomarkers of oxidative stress in affected queens.

Traditionally, surgical ovariohysterectomy was considered the elective treatment for pyometra [17]; however, since this technique results in permanent sterility and has some anesthesiologic risks [9], several alternative pharmacological treatments have been proposed in the last decade.

In this regard, prostaglandins (PGF2α) and its synthetic analogue cloprostenol have been used for the treatment of open cervix pyometra [18] as they improve myometrial contractility and, in some cases, can induce luteolysis (mainly in livestock). However, use in cats has often shown side effects such as nausea, prostration, vomiting and diarrhea within minutes after injection [19].

Aglepristone is a competitive progesterone antagonist: in cats, it binds progesterone receptors with an affinity nine times higher than the endogenous hormone, decreasing intrauterine progesterone concentration [20] and thus promoting the recovery of uterine contractility. For the feline species, the recommended dose is 15 mg/kg [8,21], due to different pharmacokinetics compared to dogs [22].

Protocols combining aglepristone and cloprostenol have been proposed [8]: aglepristone is administered at Day 1 (D1), D2, D7 and 14 [8,20], whereas cloprostenol treatment should be started at least 48 h after the first aglepristone administration (days 3 to 5) [8].

To the best of the authors’ knowledge, there is only one report of the use of aglepristone alone in the feline species with the classical administration scheme (D1, 2, 7 and 14) and the classical dose of 10 mg/mL [7]; instead, for the medical treatment of canine pyometra, Contri et al. proposed a modified protocol, administrating the aglepristone alone, with a modified scheme (D0, D2, D5, D8) [23].

In this background, the present study aims to report a case series of cats successfully treated for pyometra using the modified administration protocol proposed by Contri [23], using the manufacturer’s recommended dose of 15 mg/kg [21,22] for future routine application in queens intended for breeding with a high anesthesiologic risk or with a high sensitivity to prostaglandins side effects. Furthermore, the report aims to describe a clinical pattern different from the symptoms typically reported in the literature for feline pyometra, but frequently found in breeding subjects.

## 2. Case Description

A total of five intact queens were referred to two Italian Veterinary care centers: “Clinica Veterinaria PalermoVet”, located in Palermo, Sicily, and “VRC—Centro di Referenza Veterinario”, located in Turin, Piedmont. In all cases, the owners provided signed informed consent for clinical evaluation, diagnostic exams, and administration of therapies. All investigations were conducted in accordance with current Italian animal welfare laws.

The patients belonged to different breeds and were aged 9 to 23 month-old. For each animal, a complete anamnesis was recorded (Table 1): Patient 1 had a previous history of renal damage, while the others were all breeding subjects, recently mated.

All animals underwent a general clinical evaluation and ultrasonographic (US) examination with a Logiq P6 Pro sonographic system (GE Healthcare, Milan Italy) and a 7.5 MHz Linear probe (11L, GE Healthcare, Milan, Italy) (Patients 1 and 2) or with a MyLab 50 sonographic system (Esaote, Genoa, Italy) and a linear 12 MHz probe (Esaote, Genoa, Italy) (Patients 3 to 5). In addition, a blood sample was collected for a complete hematological and biochemical evaluation and for progesterone analysis by laser-induced fluorescence (Progesterone Speed Test; Speed Reader, Virbac, Carros, Italy). Serum progesterone higher than 2 ng/mL was considered indicative of a diestrus phase.

In all cases, a diagnosis of pyometra was made and medical treatment was chosen for its resolution: subcutaneous injections of 15 mg/kg [8,20,21,22] of aglepristone (Alizine^®^; Virbac, Carros; France) were performed at Day 0 (D0), D2, D5, and D8 after diagnosis, as described by Contri and colleagues [23].

Antibiotic therapy was associated, administrating 3 mg/kg of marbofloxacin (Marbocyl^®^, Vetoquinol S.r.l., Bertinoro FC, Italy) from D0 to D10 to limit the infection in progress, preventing worsening of clinical conditions.

Subsequent ultrasonographic evaluations were performed contextually to aglepristone administration to verify uterine involution and restore normal conditions.

In case of onset of estrus, resulting from aglepristone administration [2], the animals underwent the administration of 4 mg/kg of melatonin (Melamil^®^, Humana, Milano, Italy) for 7 days [24] after the first behavioral signs of heat.

## 3. Outcome and Follow-Up

Overall, all cases occurred between March and July; at admission, all patients showed mild clinical signs, and neither hospitalization nor supportive therapy was necessary; only in one case (Patient 1) clinical evaluation revealed hyperthermia and dysorexia. Vulvar discharge was present in two subjects (Patient 2, Patient 5).

Hematological and biochemical exams showed no alterations, except for Patient 1, in which there was an increase in azotemia and creatininemia values, suggesting an acute kidney injury, related to its previous medical history.

Plasma progesterone concentrations were always indicative of diestrus, ranging from 2.8 to 15.91 ng/ mL (8.04 ± 6.08 ng/mL).

US evaluations at time of admission revealed a mild to significant uterine enlargement, due to a pathological collection in the lumen, filled with non-homogeneous material, characterized by hypo-anechoic appearance. Uterine walls appeared outstretched, without any sign of CEH. The maximum uterine diameter recorded was 2.35 cm in patient 2 (Figure 1a).

After the beginning of aglepristone and antibiotic therapy, in all patients a mild vulvar discharge was detected in 24 h. No adverse signs were recorded during the treatment.

Ultrasonography performed at D5 confirmed, for all patients, a significant decrease in uterine size up to a normal appearance, recorded at D10 US check (Figure 2).

Ultrasonographic findings are briefly reported in Table 2.

In all treated animals, a new estrous cycle occurred from 6 to 11 days after the start of the aglepristone administration. The consequent oral administration of melatonin for 7 days proved to be useful to temporarily suspend regular cyclicality.

At the end of the therapies, Patient 1, not intended for reproduction, underwent a pharmacological estrus suppression by the application of a subcutaneous implant of a GnRH agonist (Suprerolin^®^ 4.7 mg, Virbac, Carros, France) [25], while the other four cats were re-assigned to a reproductive career, and mated at the first useful estrus. A total of 3 out of 4 mated cats became pregnant, giving birth to live and viable kittens. In the non-pregnant patient, new mating was not performed by breeder’s choice.

## 4. Discussion

Pyometra is a potentially life-threatening uterine inflammation affecting intact female cats [8,10]. Although most authors report a higher incidence in animals over 7–10 years old [1,6,7,9]. There are also some reports of cases in younger animals [5]. The authors agree with the latter, as the treated patients were less than two years old. This finding may be justified as patients were breeding subjects, and had a history of recent sterile mating, which is one of the predisposing factors for pyometra [1].

Regarding the clinical presentation of pyometra, both the authors’ experience and the cases presented confirm the evidence of scarce clinical signs in cats affected by pyometra [8], especially in young animals [5]: in this work only two patients showed vulvar discharge, and one mild systemic signs (hyperthermia and dysorexia). The authors assume that the young age (less than 2 years old) of the patients may be related to this and, effectively, Nak and collaborators reported an age range of 2–13 years for patients showing clinical signs of pyometra [7], similar to Mitacek and colleagues (2–6 years old range) [18]. Moreover, no hematological and biochemical alterations were found in the present report, differently from the above cited paper [7]. In the authors’ experience, the attention of the owners for breeding animals can allow an early diagnosis, avoiding a worsening of the clinical conditions.

Regarding the ultrasonographic (US) appearance, it is interesting to note that the examination performed did not reveal any alterations of the uterine wall suggestive of cystic endometrial hyperplasia (CEH), as reported in Table 2, thus supporting the thesis by De Bossechere and collaborators that pyometra can arise independently of CEH [11], especially in young subjects.

Another interesting finding is that the maximum uterine diameter observed by US exam during this clinical investigation was 2.35 cm, in agreement with recent research by Blanco and collaborators [26]: they proved that clinical signs of pyometra (vomiting, lethargy, hyperthermia) are associated with uterine diameters greater than 2.5 cm.

Ovariohysterectomy was not an acceptable solution in treated patients, as they were intended for breeding or with a high anesthetic risk due to previous renal damage [1,8,9,10,11,20]; therefore, the authors preferred an aglepristone-based medical treatment [20,27,28]. Aglepristone competes for uterine progesterone receptors, with an in vitro affinity, in cats, nine times higher than that of endogenous progesterone; its bioavailability is reduced due to particular pharmacokinetics [22], thus requiring a higher dose than the bitch [8,20,28,29]. The dosage of aglepristone used in this report was therefore increased to 15 mg/kg, according to the indications provided in the cited literature, but differently from that presented in previous reports [7] for the treatment of feline pyometra. This more appropriate dose, based on pharmacokinetic considerations [20,22], has recently also been described by Marino et al. in the treatment of feline mammary fibroadenomatosis [30].

In the cases presented, the authors chose to use the administration of aglepristone alone [7,23], in order to avoid the adverse effects of cloprostenol, applying the modified scheme proposed by Contri et al. [23] for the bitch. In the cited paper, the efficacy of the anti-progestin in monotherapy was explained with the possibility that blocking progesterone receptors enhances myometrial contractility in the presence of relatively low levels of the endogenous prostaglandin F2α [23], which is released because of the pathologic inflammatory conditions of the uterus [31,32].

In the present report, a supportive therapy was not requested, due to the good general conditions of the patients. Conversely, for dogs treated with the same protocol [23], hospitalization and supportive fluid–therapy was necessary; for the other studies consulted involving cats with pyometra, this information is not reported [7,9,18], although the poor clinical conditions of the patients suggested that support was necessary.

Regarding antibiotic treatment, a wide range of options is available: amoxicillin and clavulanic acid is the most common association [1,8,10], but also the association sulfadoxine-trimethoprim [7,17] is reported. To date, there are no reports of pyometra medical treatment not including a covering antibiotic treatment. In the authors’ experience, and probably due to the increasing prevalence of antimicrobial resistance, fluoroquinolones are more effective in treating uterine pathologies. The choice of marbofloxacin is also supported by recent publications, in which the use of marbofloxacin is described for pyometra treatment in dogs [33,34].

During the medical treatment with aglepristone, no side effects were detected in the treated cats, as also reported by Contri [23] for the dogs and by Nak [7] for the cats, unlike treatments involving the use of prostaglandins [9,18]. This finding agrees with a previous study by Fieni and colleagues [28] that proved that agleprisone induces side effects much less severe than prostaglandins when used for pregnancy termination in cats.

As proof of the effectiveness of the treatment, all patients showed a vulvar discharge 24 h from the beginning of the protocol. Contri et al. [23] also reported a vaginal discharge in 93% of treated patients 24 h after the first aglepristone administration and in 100% after 48 h. Protocols that also include the use of prostaglandins, on the other hand, show this result after about 48 h from the start of therapy in both dogs [35] and cats [18]. These results show that aglepristone alone is effective in promoting uterine emptying and increasing uterine contractility [32], in association with endogenous Pgf2α [31]. The results obtained, although partial and referring to a small group of animals in good general conditions, along with the bibliography consulted, support the author’s hypothesis on the efficacy of the modified aglepristone-only protocol for the treatment of pyometra in cats.

Consequently, complete uterine emptying has been achieved within 10 days from the start of therapy, with satisfactory results already on the fifth day. The same protocol applied on bitches [23] determined complete uterine involution within 14 days, thus suggesting that a aglepristone-modified administration scheme could be more efficient in queens than in bitches. Furthermore, in the feline species, the modified protocol results in a faster clinical resolution (10 days) compared to the classic one. In fact, Nak and collaborators [7] reported complete clinical, ultrasonographic and laboratory resolution on the 21st day post treatment.

However, it should be noted that the clinical condition and the ultrasound appearance of the uterus were generally good in all treated animals at the time of diagnosis. Further studies are required to demonstrate the effectiveness of the proposed protocol even in more severe cases of pyometra in the cat.

In the present study, the effectiveness of the modified protocol for aglepristone in treating pyometra was 100%. This data agree with Contri [23], who reported the same result in the application of the modified protocol on the bitch. Conversely, the effectiveness of the classic treatment, without the association with cloprostenol, is about 88% in dogs [23,36] and 90% in cats [7]. The slight difference compared to Nak’s research [7] can be explained not only by the different administration scheme but also by the dosage, adapted to the different pharmacokinetics and consequent bioavailability in the feline species, compared to dogs [8,20,22].

On the other hand, the effectiveness of the modified protocol is comparable to the protocols that provide for the association of aglepristone and cloprostenol, both for dogs and cats [8,35], but without the above-cited side effects [18,28].

These findings, with the support of the consulted literature, support the validity of the protocol proposed as an alternative to those commonly used up to now.

In all treated animals, a new estrus cycle occurred between 6 and 11 days from the beginning of the treatment, therefore before the end of the antibiotic and anti-progestin therapy. Similar data are reported by García Mitacek and collaborators [37], according to which progesterone remains high for 5–8 days after aglepristone-induced abortus, then a new estrus cycle can arise. In the present study, progesterone serum concentration after treatment was not detected, but clinical signs of estrus allowed the authors to hypothesize a similar trend for the involved patients. These findings can be justified by the hypothesis by Galac and collaborators [38] that aglepristone could have a central effect of the anti-progestagen on the hypothalamic receptors, resulting in a reduced inter-estrus, as confirmed by Contri [23 contri] for the dog.

The authors’ decision to temporarily suppress estrus cycles was made to allow the uterus to restore completely, and the patients to complete antibiotic ad anti-progestin therapy.

Several pharmacological treatments have been proposed for estrus prevention in cats: GnRH agonists [25] and melatonin [24] are the most common, with a few side effects; melatonin can be administered orally or by subcutaneous implants [24,39].

The authors preferred to use melatonin by oral administration as it determines rapidly reversible effects [39]. In this way, at the end of the medical therapy for pyometra, the patients quickly returned to their reproductive career, except for Patient 1, which was then subjected to prolonged estrus suppression by deslorelin implant [25].

Regarding fertility after treatment, in the present report, 3 (Patient 2, 3 and 4) of 4 mated queens (Patients 2, 3, 4 and 5) became pregnant at the first mating after treatment (75% pregnancy rate). In dogs, the modified aglepristone protocol resulted in a pregnancy rate of about 80% [23], while Nak [7] reported for cats treated with the classical aglepristone-only protocol that all mated cats (2 up to 9 treated animals) became pregnant. The results presented are also better than those obtained after cloprostenol administration [18].

Finally, a long-term follow-up highlighted the absence of pyometra recrudescence during the two years after treatment, both for dogs treated with modified aglepristone protocol [23] and for cats that underwent the classical aglepristone administration scheme [7]. Further studies are required to evaluate the long-term prevention of recrudescence in cats treated with the proposed protocol, but the authors hypothesize that the results may be similar to the above-cited literature.

## 5. Conclusion

The clinical pattern reported in the present paper is milder than that generally reported in the bibliography. In the authors’ experience, this clinical presentation is frequently found in breeding subjects due to correct management that allows a rapid diagnosis and effective clinical treatment.

Moreover, the proposed modified administration protocol with aglepristone proved to be useful and promising, whereas its validity needs to be tested on a larger population with a more severe clinical presentation, for future routine application for the medical treatment of feline pyometra, in queens intended for breeding, with a high anesthesiologic risk or with a high sensitivity to prostaglandins side effects.

## Figures and Tables

**Figure 1 vetsci-09-00517-f001:**
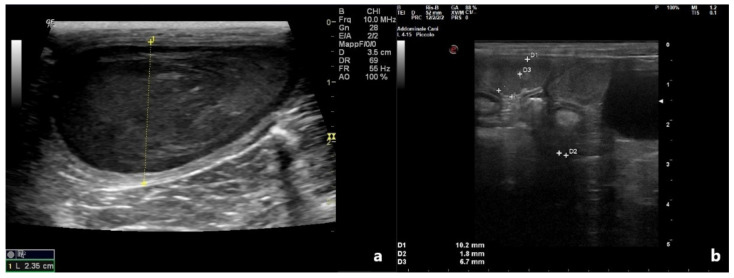
Ultrasonographic appearance of affected uterus, at the time of diagnosis. Uterine content has a heterogeneous echotexture. (**a**) Patient 2; (**b**) Patient 4.

**Figure 2 vetsci-09-00517-f002:**
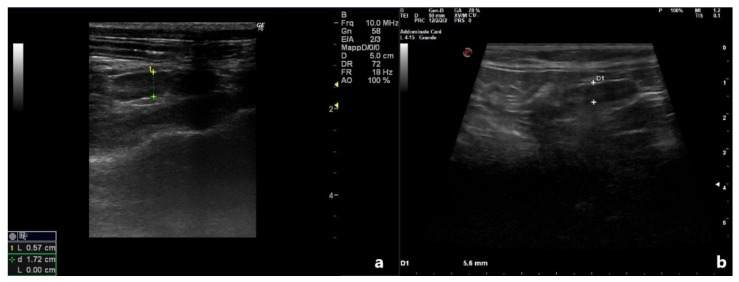
Ultrasonographic appearance of the same uterus reported in Image 1, 10 days after treatment. (**a**) Patient 2; (**b**) Patient 4.

**Table 1 vetsci-09-00517-t001:** Signalment and history of animals included in the case series.

Animals	Month of Referral	Age	Breed	Referred For	History
Patient 1	March	9 months	Domestic Short Air	General illness	Renal damage
Patient 2	May	16 months	Main Coon	Suspicion of vaginitis	Recent mating
Patient 3	April	23 months	Siamese	Pregnancy diagnosis	Recent mating
Patient 4	May	13 months	Bengal	Pregnancy diagnosis	Recent mating
Patient 5	July	20 months	Main Coon	Vaginal Discharge	Recent mating

**Table 2 vetsci-09-00517-t002:** Ultrasonographic findings of animals included in the case series. D0 refers to the day of diagnosis and start of aglepristone treatment. D10 refers to the last day of antibiotic treatment, corresponding to the complete ultrasonographic recovery.

	Uterine Diameter (cm)		Uterine Wall (cm)		Luminal Collection	
	D0	D10	D0	D10	D0	D10
Patient 1	0.75	0.49	0.2	0.05	Hypoecoic	No
Patient 2	2.35	0.57	0.3	0.10	Heterogeneous	No
Patient 3	0.41	0.23	0.1	0.1	Anechoic	No
Patient 4	1.02	0.56	0.18	0.10	Anechoic	No
Patient 5	0.80	0.37	0.15	0.10	Hypoecoic	No

## Data Availability

The data presented in this study are available on request from the corresponding author.

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
