# Peer review of "Effectiveness of a Modified Administration Protocol for the Medical Treatment of Feline Pyometra"

_vetsci, 2022, doi:10.3390/vetsci9100517_

Round 1
Reviewer 1 Report (Previous Reviewer 3)
Authors have revised the manuscript carefully. They have considered all points I raised. The overall quality of the manuscript has been improved.
In the current case study, the authors tried to provide a case series presentation of some cats with pyometra treated using a modified aglepristone protocol. The study is proper to be presented in the present form as a case report study.
The study is well organized and presented.
Despite this medication being used before in dogs, I think that this work will provide new knowledge about using aglepristone in cats. This will be interesting to the readers.
Finally, the methodology and the conclusion are consistent with the study objectives.
Author Response
Authors have revised the manuscript carefully. They have considered all points I raised. The overall quality of the manuscript has been improved.
In the current case study, the authors tried to provide a case series presentation of some cats with pyometra treated using a modified aglepristone protocol. The study is proper to be presented in the present form as a case report study.
The study is well organized and presented.
Despite this medication being used before in dogs, I think that this work will provide new knowledge about using aglepristone in cats. This will be interesting to the readers.
Finally, the methodology and the conclusion are consistent with the study objectives.
We would like to thank the Reviewer again for his suggestions, which were extremely helpful in improving the manuscript.

Reviewer 2 Report (Previous Reviewer 1)
Review Report
- A summary
The case series was aimed at reporting the effectiveness of a modified treatment protocol for feline pyometra.
- Broad comments
The case series seems to be carefully reported. However, some more information about the individuals is needed. Please see the specific comments.
- Specific comments
To clarify the purpose of this case series, I would suggest explaining your diagnostic strategies more systematically.
Page 3 line 108: The time (month) of the referral is an important piece of information that should be addressed.
Page 4 line 132: Please explain what the indication for antibiotic use is and why it is important to include this in the treatment protocol. Is that necessary in patients with relatively good general status without any signs of general infection?
Page 4 line 141: I would suggest mentioning which clinical signs you recorded in different cases.
Page 4 line 152: Did you measure the thickness of the uterine wall in different patients?
Page 4 line 153: I would suggest reporting the ultrasound findings (measurements or ultrasound images) in other patients as well. It could be at least included in supplementary materials.
Page 9 lines 332-333: Reformulation is needed. To make this statement, a controlled experiment with control groups is needed. I would therefore suggest reformulating this statement.
Author Response
Page 3 line 108: The time (month) of the referral is an important piece of information that should be addressed.
All patients were referred between March and July. For a better description of the cases, another column has been added to Table 1, with specific information. Moreover, in line 121 the following statement has been added “cases occurred between March and July”
- Page 4 line 132: Please explain what the indication for antibiotic use is and why it is important to include this in the treatment protocol. Is that necessary in patients with relatively good general status without any signs of general infection?
Currently, to the Authors’ knowledge, there is no report of medical treatment of pyometra (both for dogs and cats) not including antibiotic treatment. Recent communications suggest not using an antibiotic only in cases of surgical resolution. In cases of medical treatment, instead, the source of infection does not suddenly disappear but is gradually reduced until it is eliminated by uterine clearance. For this reason, a covering antibiotic treatment should be administered to counteract bacterial activity and prevent the worsening of the clinical conditions. In the discussion section, the following statements have been added:
“Antibiotic therapy was associated, administrating 3 mg/kg of marbofloxacin (Marbocyl®; Vetoquinol S.r.l., Italy) from D0 to D10, to limit the infection in progress, preventing worsening of clinical conditions.” (Lines 112-114)
“At the current date, there are no reports of pyometra medical treatment not including a covering antibiotic treatment” (lines 213-214).
- Page 4 line 141: I would suggest mentioning which clinical signs you recorded in different cases.
Clinical signs were mild to absent in all treated animals, as stated in lines 121-124 (outcome and follow up).
The Discussion section has been modified as follows: “Regarding the clinical presentation of pyometra, the Authors' experience, and the cases presented, confirm the evidence of scarce clinical signs in cats affected by pyometra [8], especially in young animals [5]: in this work only two patients showed vulvar discharge, and one mild systemic signs (hyperthermia and dysorexia). The authors assume that” (Lines 168-172)
- Page 4 line 152: Did you measure the thickness of the uterine wall in different patients?
- Page 4 line 153: I would suggest reporting the ultrasound findings (measurements or ultrasound images) in other patients as well. It could be at least included in supplementary materials.
A summary table of the ultrasound characteristics has been added (Table 2, Line 148). Moreover, the Discussion section has been modified as follows. “It is interesting to note that the examination performed did not reveal any alterations of the uterine wall suggestive of cystic endometrial hyperplasia (CEH), as reported in Table 2” (Lines 179-181).
Figures 1 and 2 have been modified, as well (lines 137 and 147).
- Page 9 lines 332-333: Reformulation is needed. To make this statement, a controlled experiment with control groups is needed. I would therefore suggest reformulating this statement.
Conclusions have been modified as follows: “Moreover, the proposed modified administration protocol with aglepristone proved to be useful and promising, whereas its validity needs to be tested on a larger population with a more severe clinical presentation, for future routine application for the medical treatment of feline pyometra, in queens intended for breeding, with a high anesthesiologic risk or with a high sensitivity to prostaglandins side effects.” (Lines 297-301)

Reviewer 3 Report (New Reviewer)
This article describes a shorter protocol for medical treatment of pyometra in cats. Which is of interest to the field, especially to breeders, however the conclusion as it is stated now is too strong. In this study, a small number ofh only very young cats were included, only cats without signs of CEH and the follow-up period was too short to check for recidives, so I believe that it's too optimistic to state that this is a safe and effective protocol for the whole cat population.
Line 121: did you perform an ultrasound examination on patiënt 5 as well?
Line 133: Why did you choose marbofloxacine? This should not be the first choice antibiotic in these cases. It's a third generation fluoroquinolone, which should only be used after sensitivity testing. I believe we should not promote this as common practice in the light of antimicrobial resistance.
Line 137: It's not clear to me when the melatonin treatment started in respect to the alizin injections, and when the first oestrus occured in the light of the melatonin treatment. Do I understand correctly that the queens showed oestrus signs during melatonin treatment?
Line 177: I see her that the fertility rate after alizin injections is 3/4, although in the abstract it is mentioned 'normal fertility was preserved in all breeding subjects'. Is there any information available on the queen that was not pregnant after the first mating? was she pregnant after a second or third mating?
The discussion contains a lot of information that is already available in the introduction. There is too much repetition.
Author Response
This article describes a shorter protocol for medical treatment of pyometra in cats. Which is of interest to the field, especially to breeders, however the conclusion as it is stated now is too strong. In this study, a small number of only very young cats were included, only cats without signs of CEH and the follow-up period were too short to check for recidives, so I believe that it's too optimistic to state that this is a safe and effective protocol for the whole cat population.
Conclusions, and related statement in the abstract, have been modified as follows:
“..the proposed modified protocol proved useful and promising for the medical treatment of pyometra in cats..” (Abstract, lines 32-33)
“Moreover, the proposed modified administration protocol with aglepristone proved to be useful and promising, whereas its validity needs to be tested on a larger population with a more severe clinical presentation, for future routine application for the medical treatment of feline pyometra, in queens intended for breeding, with a high anaesthesiologic risk or with a high sensitivity to prostaglandins side effects.” (Lines 297-301)
- Line 121: did you perform an ultrasound examination on patiënt 5 as well?
We apologize for the inconvenience caused by a typing mistake when rewriting the manuscript. All patients underwent US exams. The manuscript has been corrected as follows “(Patients 3 to 5)” in line 104.
Moreover, a table resuming ultrasonographic features has been added (table 2 – line 148)
- Line 133: Why did you choose marbofloxacine? This should not be the first-choice antibiotic in these cases. It's a third-generation fluoroquinolone, which should only be used after sensitivity testing. I believe we should not promote this as common practice in the light of antimicrobial resistance.
The Authors fully understand and agree with the problem of antimicrobial resistance. Unfortunately, the cases presented are proof of this: in fact, involved patients have a known resistance (not reported in the manuscript) to the main classes of broad-spectrum antibiotics, such as the amoxicillin-clavulanic acid association. So, with the support of cited bibliography (Ros, 2014; Melandri 2019) and based on clinical experience that fewer and fewer subjects respond positively to a first-choice antibiotic treatment (thus having to pass, after an antibiogram, to another pharmacological class), the Authors chose to start treatment with marbofloxacin directly, to avoid waiting for a complete antibiogram and then having to change therapy thus prolonging the time of antibiotic administration.
- Line 137: It's not clear to me when the melatonin treatment started in respect to the alizin injections, and when the first oestrus occurred in the light of the melatonin treatment. Do I understand correctly that the queens showed oestrus signs during melatonin treatment?
For melatonin administration, no default day has been chosen for the start of treatment, but it has been administered only if animals showed estrus signs. In all considered cases, a new estrus cycle appeared from 6 to 11 days after the first aglepristone administration consequently, animals were treated with melatonin for 7 days to promote short-term suppression of the estrous cycle.
For a better understanding, manuscript has been modified as follows:
“In case of onset of estrus, resulting from aglepristone administration, the animals underwent the administration of 4 mg/kg of melatonin (Melamil®; Humana, Milano, Italy) for 7 days after the first behavioral signs of heat.” (Lines 117-119)
“In all treated animals, a new estrous cycle occurred from 6 to 11 days after the start of the aglepristone administration. The consequent oral administration of melatonin for 7 days proved useful to temporarily suspend regular cyclicality.” (Lines 152-154)
- Line 177: I see her that the fertility rate after alizin injections is 3/4, although in the abstract it is mentioned 'normal fertility was preserved in all breeding subjects'. Is there any information available on the queen that was not pregnant after the first mating? was she pregnant after a second or third mating?
Unfortunately, as of the current date, we do not have this information as, by choice of the breeder, Patient 5 has not yet been mated again. However, oestrus cycles are regular, and, at the moment, she has not shown recurrence of pyometra.
Abstract has been modified as follows: “After treatment, 3 queens had an uneventful pregnancy.” (Lines 30-31)
Outcome section has been modified as follows: “In the non-pregnant patient, a new mating was not performed, by breeder’s choice.” (Lines 159-160)
Discussion section has been modified as follows: “Regarding fertility after treatment, in the present report 3 (Patient 2, 3 and 4) of 4 mated queens (Patients 2, 3, 4 and 5) became pregnant at the first mating after treatment (75% pregnancy rate).” (Lines 280-281)
- The discussion contains a lot of information that is already available in the introduction. There is too much repetition.
A reformulation has been made

Round 2
Reviewer 2 Report (Previous Reviewer 1)
Authors have responded to my comments adequately and changes are acceptable.
This manuscript is a resubmission of an earlier submission. The following is a list of the peer review reports and author responses from that submission.
Round 1
Reviewer 1 Report
Review Report
- A brief summary
This study aimed at to investigate the effectiveness of a modified treatment protocol for feline pyometra. Only four affected queens were enrolled in this study and all of them received the same treatment.
- Broad comments
The methodological approaches and derived conclusion in this study are not corresponding. The design of this study is not corresponding to an original research study. The lack of a fundamental research structure is clearly observable. Although it is reported that all four candidates were re-assigned to a reproductive career after treatment, the effectiveness of this protocol can’t be approved with this small sample size and lack of controls. This study is better suited to be presented as a case report. I would therefore strongly suggest reformulating and converting the current manuscript into a case report without drawing a research-based conclusion. This is provided that the journal accepts such a strategy.
Reviewer 2 Report
Dear Editor,
After careful examination of the submitted manuscript, unfortunately I must recommend rejection of this manuscript due to the very limited scientific value of the presented data.
Authors present results of treatment that was previously described by other ( cited in the text) authors and main modification is higher doses of Algepristone. Another problem is that they present the results obtained on only four patients, which is in my opinion insufficient to make a serious conclusion. In this context high effectiveness cannot surprise.
Regarding the construction of manuscript- similar information are repeated in the introduction and discussion.
The use of melatonin is according to the authors recommended element of a treatment but the article seems to describe only just a treatment with the use of algepristone. If results (long effect in the context of fertility) are evaluated after melatonin administration it is not treatment with only algepristone- but with algepristone and melatonin.
Once again I will say that first of all there is nothing new in his protocol since Nack et al ( 2009) also used treatment of algepristone without prostaglandins obtaining the same very good results but on the much bigger number of patients. The difference in dosage is not so important if lower doses used by Nack was sufficient and effective.
I would propose the authors to gather a bigger number of cases and compare the longitudinal effects of treatment with and without use of melatonin, which in my opinion could be a new and interesting element in the proposed protocol.
Reviewer 3 Report
In the present study, the authors tried to assess the efficacy of a modified aglepristone protocol for treatment of pyometra in cats. However, there are many limitation factors:
1. The number of the animals is very limited (4 cases only) that is not enough to evaluate the modified protocol. I recommend resubmission of this paper as a case series or as a short communication.
2. The authors preferred to use the higher dose of aglepristone (15mg/kg) rather than the usually used dose in cats (10mg/kg). Despite their explanation in the discussion section, this point is a subject for further investigation.
3. The post treatment long term evaluation of the recurrence rate was not presented which is also subject for further investigation.
4. In the abstract (lines 16-17), the authors mentioned that the aim of their work was to evaluate the efficacy of aglepristone-only modified protocol for treatment of pyometra in cats. However, other drugs as antibiotics (Marbofloxacin) and melatonin. this point is not clear enough.